# Change to a Plant-Based Diet Has No Effect on Strength Performance in Trained Persons in the First 8 Weeks—A 16-Week Controlled Pilot Study

**DOI:** 10.3390/ijerph20031856

**Published:** 2023-01-19

**Authors:** Eduard Isenmann, Laura Eggers, Tim Havers, Jan Schalla, Alessio Lesch, Stephan Geisler

**Affiliations:** 1Department of Fitness and Health, IST Hochschule of Applied Sciences Düsseldorf, 40233 Düsseldorf, Germany; 2Department of Molecular and Cellular Sports Medicine, Institute of Cardiovascular Research and Sports Medicine, German Sport University Cologne, 50933 Cologne, Germany

**Keywords:** plant-based diet, vegan diet, body composition, performance

## Abstract

Over the past few years, the number of people who have avoided animal products has been rising steadily. A plant-based diet is associated with a healthier lifestyle and has positive effects on various diseases. More and more healthy active people and performance-orientated athletes are giving up animal products for various reasons, such as for an improved performance or faster regeneration. However, the data in this context are limited. This study aimed to obtain initial findings on the influence of a diet change to veganism on the performance of strength-trained individuals. For this study, a total of 15 omnivorous individuals were recruited. They documented their dietary food intakes over 16 weeks. Every four weeks, the strength performance was tested via a leg press and bench press. In the first 8 weeks, the participants maintained their omnivorous diet, followed by 8 weeks of a vegan dietary phase. In total, 10 subjects participated successfully, and their data were part of the statistical analyses. There was no difference in the absolute and relative strength performance for the leg and bench press after changing to a vegan diet. For the total calorie intake and carbohydrates, only a small treatment effect, but no time effect, was observed. However, for the protein intake, a time and group effect were detected. In addition, the relative protein intake decreased significantly and was lower than the current recommendations for athletes. The results demonstrate that a change to a vegan diet has no beneficial nor negative effect on the strength performance when the total calorie intake and carbohydrate content are covered in the first 8 weeks.

## 1. Introduction

An increasing number of people are reducing their consumption of animal-based foods or abstaining completely from eating these products [1]. A complete renunciation of animal-based foods is referred to as a plant-based or vegan diet. Over the last 25 years, for example, the number of vegan people in the USA has risen from around 0.3–0.5 to 2.5–6 million [1]. A similar development was observed in other countries such as the UK, Germany, and Australia [1]. The reasons for this are very different. In addition to moral and sustainable aspects, health factors are also among the motives [2]. Recent reviews have shown that the increased consumption of animal-based foods, especially red meat and highly processed products, is related to higher risks of various types of cancer [3,4,5,6]. Consequently, a vegetarian or vegan diet is often associated with a healthy diet. So far, a vegan diet has only been scientifically studied regarding the health aspects. Reviews have shown that a vegan diet has a positive influence on the cardio-metabolic risk profile [7], significantly lower C-reactive protein levels [8] and reduces the level of total cholesterol in the blood [9]. A vegan diet also showed anti-inflammatory effects in several studies, which are relevant in the context of lifestyle diseases [10,11,12]. Consequently, a positive influence on the development of obesity, hypertension, diabetes, and cardiovascular mortality has also been shown [13].

However, in addition to these positive effects on health, there are also several risks of a vegan diet. This diet is often associated with an undersupply of macro- and micro-nutrients. Compared to omnivorous diets with lower energy intakes [7], a vegan diet is also often associated with a deficit in the protein supply. Bakaloudi et al. [14] found in their review that the protein intake in a vegan diet is lower compared to other diets, even though there were only slight deviations from the WHOs recommendation (13–15% of the total energy intake by protein at a 15% recommendation). In terms of micronutrients, the intakes of vitamin B12, zinc, calcium, and selenium deviated from the WHOs recommendation in this review. No disadvantages were observed for vitamins A, B1, B6, C, E, iron, phosphorus, magnesium, copper, and folic acid. Nevertheless, it is known that a plant-based diet, with a high content of legumes and whole-grains, can influence the bioavailability of iron and zinc [15]. However, a varied plant-based diet in industrialized countries has not been shown to have adverse health effects due to a lower iron and zinc intake [15]. A vegan diet also entailed a lower glycemic load [14].

So far, a plant-based diet has not been sufficiently studied in sports science. A few existing studies in this context showed that a vegan diet could be associated with a positive health status [16]. Furthermore, it could be shown for endurance athletes (recreational runners to ultramarathon runners) that there is a positive influence on the quality of life [17]. With a well-planned and balanced vegan diet and the supplementation of vitamin B12, deficiencies of the critical nutrients vitamin B12 and iron can be avoided in athletes [18]. The diet should be rich in wholegrains, legumes, nuts, seeds, dried fruits, iron-fortified cereals, and green leafy vegetables to ensure a sufficient iron intake and should be supervised by a nutritionist [19]. In addition to the health aspects, questions referring to the physical and athletic performance are essential for athletes in competitive and elite sports. A study of young healthy women showed that a vegan diet had no disadvantages for a maximum oxygen uptake, with submaximal endurance increased for the vegan participants [20]. Nebl et al. [21] investigated the effect of a vegan diet on the performance of recreational runners. They found no difference in the maximal performance compared to vegetarians or omnivores. Regarding strength training and strength capacity, the data are also insufficient. Initial studies could not find a clear difference between an omnivorous and a vegan diet in terms of strength endurance after a four-week dietary change [22]. The authors found a slight effect on the strength endurance in the deadlift exercise during a vegan diet, while no beneficial effects were detected in the squat exercise and a specific CrossFit workout. Furthermore, recent research shows no difference in the performance between male long-term vegans and omnivorous athletes [23]. For muscular adaptions, Hevia-Larraín et al. [24] found no difference in muscle growth between vegans and omnivores after 12 weeks of strength training, provided that the protein intake was equal. However, there are no available data on the effect of a change to a vegan diet on the strength capacity.

Based on the lack of studies on the topic of strength training, a 16-week intervention was conducted with the aim of investigating the influence of changing to a vegan diet on omnivorous athletes. The aim of this study is to highlight the influences of different diets on the lower and upper limb strength.

## 2. Methods

### 2.1. Participants

Young healthy volunteers aged 18–35 years with at least six months of strength training experience at a local gym (Häsler Vital Center, Horb am Neckar, Germany) were recruited. Another requirement was that both the leg press and bench press were a regular part of the training routine. For the bench press exercise, this was ensured via the classification according to Santos Jr. The participants had to be at least at an intermediate level [25]. A total of 15 (m = 5; f = 10) people participated in the study. All participants engaged in strength training at least twice a week. An increase in the training frequency as well as a change in the target (maximum strength or hypertrophy) was prohibited during the entire intervention period. The training frequency was monitored throughout the entire period.

### 2.2. Study Design

A three-phase study design was created. Initially, all participants were given an explanation of the vegan and omnivorous diet. In the first two-week phase, there was a familiarization of the documentation of the diet, as well as the performance of the maximum strength tests. Subsequently, strength tests were performed and anthropometric data were collected. This was followed by an 8-week control phase with a normal diet (omnivore) (phase 2). In phase 3, all participants changed to a vegan diet for 8 weeks. Three days before the start of phase 3, all participants were informed again about a vegan lifestyle by a sports nutrition coach. In phases 2 and 3, both the strength capacity and anthropometric data were measured every 4 weeks. The nutrition diary was checked every day by an employed sports nutritionist from the gym. All participants were supervised by this person during the entire intervention period. In addition, this person examined the training habits as well as the training plans of the participants. The training habits were not influenced during this time and had to be maintained in all phases. The complete study design is shown in Figure 1.

### 2.3. Dietary Strategies and Documentation

#### 2.3.1. Omnivore Diet

During phase 2, the existing dietary habits should be maintained and not explicitly adapted. All participants followed an omnivorous diet at the beginning of the study. During the omnivorous phases (phase 1 and 2), all animal- and plant-based foods were allowed to be consumed. There were no restrictions on food choices. People who were already following a vegan lifestyle or a vegetarian diet could not participate in the study. Furthermore, there were no guidelines on the macronutrient distribution.

#### 2.3.2. Vegan Diet

In phase 3, only plant-based food was allowed. All meat and dairy products, as well as any other animal-based products such as eggs or honey, were prohibited during the entire period. In addition, food in which animal products were used for production could not be used. If participants unintentionally consumed foods with ingredients of animal origin, they were not excluded from the study. Nevertheless, before the start of the third phase, the participants were instructed regarding the permitted foods (T1b). Furthermore, the participants could consult the nutrition coach at any time. During phase 3, the energy expenditure and macronutrient distribution were noted exclusively. The participants were not given any instructions to achieve the same consumption or macronutrient distribution as in phase 2. The sports nutritionist nevertheless checked the total number of calories and the macronutrient distribution weekly so that no deficiency symptoms could occur.

#### 2.3.3. Nutrition Documentation

During the entire 16-week period, the diet was monitored using the Food Diary Food Database (FDDB) Extender (Food Database GmbH, 28217 Bremen, Germany). All the food and beverages consumed during the day had to be documented. The FDDB Extender is an established method for validly recording the macronutrient distribution [26]. The FDDB Extender has therefore been used in previous nutrition interventions [27]. In addition, the basal metabolic rate would be calculated using the Harris–Benedict formula and multiplied by the physical activity level factor to determine the participants’ daily energy expenditure. The participants were informed about their individual energy consumption before phase 1, but this only served as a check. If this did not differ from the self-documented energy intake, it was not taken into account. Regarding the macronutrient distribution, no guidelines for carbohydrates, proteins, and fats was given in any phase. The macronutrient distribution of phase 2 and 3 was only retrospectively compared with the current recommendations of the International Society of Sports Nutrition. In addition, the consumption of dietary supplements was prohibited during the entire period. If individual participants consumed dietary supplements before the start of the intervention, they were also allowed to use them during the intervention. In phase 3, however, food supplements such as omega 3 capsules or protein powder were not allowed to consist of animal products and had to be replaced by vegan products. At the beginning of the study or during the transition from phase 2 to 3, no food supplements were allowed to be used to compensate for possible deficits.

##### Harris–Benedict Formula

Male:Basic metabolic rate [kcal/24 h] = 66.47 + (13.7 × body weight [kg]) + (5 × height [cm]) − (6.8 × age [years])(1)

Female:Basic metabolic rate [kcal/24 h] = 655.1 + (9.6 × body weight [kg]) + (1.8 × height [cm]) − (4.7 × age [years])(2)

### 2.4. Measurements

#### 2.4.1. One Repetition Maximum Test

To test the maximum strength of the participants, one repetition maximum test (1-RM) was performed in the leg press and bench press exercises. The test was always performed at the same gym, with the same equipment, at the same time, and on the same day of the week. The test protocol was executed based on the guidelines of the NSCA [28]. After a 5 min individual warm-up, a specific warm-up with a predefined intensity and number of repetitions followed. At 50% of the estimated 1 RM, the test persons had to complete 10 repetitions in the first set. After a 3 min break, a second set followed (80% estimated 1 RM, 3 repetitions). This program applies to both exercises. Subsequently, a minimum of 3 and a maximum of 6 1 RM tests per exercise were performed. If a weight was successfully mastered in the respective exercise, the weight was increased in the next set (2.5 kg for the bench press and 5 kg for the leg press). If a weight was no longer successfully mastered after the increase, a final test followed for which the weight was reduced by half of the increase. A 4 min break was taken between each 1 RM attempt.

#### 2.4.2. Anthropometric Parameters

At the beginning of the study, the body weight (Seca803; Seca GmbH & Co. KG, Hamburg, Germany) and height was recorded using a wall-mounted stadiometer (Seca206; Seca GmbH & Co. KG, Hamburg, Germany). Based on this, the body mass index (BMI) was also determined. The weight was checked every 4 weeks and the BMI was also determined.
BMI = body weight (kg)/height(m)^2^

### 2.5. Statistical Analyses

For statistical analysis, the current version of the SPSS (IBM SPSS Statistics 29.0, Ehningen, Germany) was used. All results were presented as mean values with standard deviations (mean ± SD). A linear mixed model (LME) was used to identify the time (five and 16-time points) and treatment effects (omnivore and vegan). For the analysis, the time and the treatment were defined as fixed effects and the individuals were considered as random effects. In addition, the maximum likelihood of the method was set with an iteration of 1000. Interaction effects could not be carried out due to the study design. The significant difference was set at *p* < 0.05. Furthermore, the effect size for significant group differences was calculated using Cohen’s d. Marginal effects were defined as up to 0.2, small effects up to 0.5, medium effects up to 0.8, and large effects from 0.8.

## 3. Results

### 3.1. Anthropometric Data and Training Frequency

A total of 10 (m = 3; f = 7) of the 15 participants completed the 16 weeks of the examination. Two participants had to leave the study in the omnivorous phase due to injury and illness, while another two participants dropped out due to family and career changes. Only one of the dropouts reported a severe decline in performance in phase 3, which led to his own decision to drop out. However, there were no health complaints. No significant time effect was detected for weight loss (*p* = 0.623) and for BMI (*p* = 0.593) over the entire period. However, a significant treatment effect was observed for both parameters (weight: *p* = 0.003 #; BMI: *p* = 0.002 #). The effect for both parameters was trivial in each case (Table 1).

### 3.2. Performance Parameter

No significant time and group difference could be determined in the absolute and relative strength capacity of the leg press. Additionally, in the bench press exercise, no time or group difference could be detected. All mean values with SD and the *p*-values are shown in Table 2.

### 3.3. Total Calorie Consumption and Macronutrient Distribution

The nutrition diaries were kept daily over a period of 16 weeks. The total calorie intake and macronutrients were summarized weekly. No significant change in self-documented total kilocalorie consumption was observed over the entire period (*p* = 0.671). Between weeks 1 and 16, on average, 103 ± 209 kcal were added (Figure 2A). However, a significant difference with a small effect was noted between the two treatments (Mean: −60.65 SD: 98.47; *p* = 0.011 #; d = 0.295).

A time trend in the self-reported carbohydrate intake was observed and a treatment difference was identified (time: *p* = 0.119; treatment: *p* = 0.001 #). The group difference can be classified as small to moderate (d = 0.492) (Figure 2B). The self-reported protein intake showed both a time (*p* = 0.022 *) and a moderate treatment difference (*p* < 0.001 #; d = 0.670). The relative protein intake also decreased over time (*p* = 0.006 *) and a moderate treatment effect is noticed (*p* = 0.001 #; d = 0.687) (Table 3). No difference for the time or treatment could be found for the fat intake (time: *p* = 0.533; treatment: *p* = 0.059) (Figure 2D).

## 4. Discussion

This pilot study aimed to obtain initial findings on the influence of a change to a plant-based diet on the strength capacity of trained individuals. At the present time, no data are available on this topic to the authors’ knowledge, especially for the period of an acute dietary change.

Based on the results, it seems that a change in diet does not influence the absolute strength capacity in the lower and upper body if the total calorie requirement and carbohydrate content are covered. However, a clear difference in the absolute and relative protein intake was detected. Additionally, a dietary change seems to have a trivial influence on the body weight and BMI during the first 8 weeks in physically active persons.

With the assumption that a plant-based diet is associated with a healthier lifestyle, more and more people, including athletes, are changing their diets and reducing or avoiding animal-based foods [29,30]. However, in this context, the health status, as well as the different lifestyles of the various populations, must be considered. Healthy and physically active individuals usually do not have health limitations or chronic inflammation such as obese individuals or type 2 diabetes patients do [10,31]. That a plant-based diet has a positive influence on these groups has already been observed in several studies [10,29]. Both the body composition and the inflammatory reactions could be reduced and thus the quality of life increased [10,29]. For people who are active in sports, a change in body weight or body composition is not urgently necessary but is usually only a personal and individual goal; this is especially true for people who exercise regularly in attempt to continuously improve their physical performance. This includes implementing all factors that can influence performance, such as training, nutrition, and regeneration, in the best possible way. Nutrition is becoming more and more important in this context. However, performance can also be impaired by malnutrition or the incorrect timing of nutritional intake [32,33]. Therefore, the primary goal of a change in diet is to ensure that performance is not negatively affected. As reported in previous studies, a vegan diet regularly leads to a lower calorie intake [7,14]. This observation was also made in this study. A significant difference between the two treatments was detected. However, the self-documented difference between the two diets was small. In the distribution of macronutrients, it was also reported that the participants would consume more carbohydrates during the period of the vegan diet than during the omnivorous phase [14]. This assumption could also be confirmed in this study. Even though only a trend can be observed over time, a group difference could also be detected. A possible explanation could be the compensation of calories from animal products by whole-grains, legumes, and seeds. They are also known to be rich in carbohydrates, which explains the slight increase in the carbohydrate content. However, this effect is also only small to moderate if referring to the treatment difference. Since the self-documented calorie intake, as well as the carbohydrate intake, were almost equal for both treatments, it might explain the maintenance of the strength parameters over time. Consequently, the calorie requirement as well as the carbohydrate content must be covered. In addition, both omnivorous and vegan diets use mainly plant-based foods to cover the carbohydrate requirement, so there is hardly any difference between the two diets. In contrast, proteins and their supply play an important role in physiological adaptation. It was found that there was a significant time and group difference in both the absolute and relative intakes. This is consistent with previous observations [14]. The relative protein intake to body weight is decisive here. The current recommendations for the protein intake of athletes and physically active persons ranged between 1.2 and 2.0 g/kg/BW [34,35]. Considering the relative protein intakes during the omnivorous and vegan phases, it is evident that during the omnivorous phase, the relative intake is within the current recommendations. By switching to a vegan diet, they are no longer within the recommended range. Consequently, if the diet is changed to plant-based without considering the relative protein intake, the physiological adaptation mechanisms, such as skeletal muscle growth, could be hindered. However, this has not yet been sufficiently investigated. In contrast, existing studies could show that with the same protein intake, in the range of the recommendations, no differences between a vegan and omnivorous diet can be observed [24]. In addition to the total protein intake, attention should also be paid to the amino acid composition in a vegan diet. It is known that plant foods, such as soy and quinoa, individually have a poorer amino acid distribution than animal foods [36]. However, this can be counteracted very well by the right combinations of plant foods. For body weight and BMI, a trivial effect was detected. Considering that only a slight reduction in the calorie intake was documented after the transition from an omnivorous to vegan diet, the effects on the body composition are understandable. A possible long-term effect of a vegan diet could be therefore a reduction in body weight and an improvement in BMI. However, this is only relevant for populations that are not at the lower ranges of a healthy body weight and BMI. Consequently, body weight, BMI, and the reduction of both is primarily related to one’s total intake. The nutrition strategy is not decisive. However, in terms of the body composition and the ratio between the fat-free mass and fat mass, the difference between the two dietary strategies can only be speculated in this context. If the relative protein intake is considered, it is conceivable that the trivial change is also due to a slight decrease in fat-free mass. However, this cannot be shown by this study but should be included in further studies.

Besides the important new findings from this study, it also has some limitations. Due to the sample size, the significance of the effects on the individual parameters is limited and must therefore be considered as a pilot project. Additionally, the data collected are mainly based on the athletes’ self-documented food diaries and probably does not reflect their actual intake. However, this bias is present in all phases of the study and in previous studies, so this can be partially neglected. Furthermore, no conclusions can be made about the influence on the fat or fat-free mass. In addition, the majority of the population was female, so in future studies, the menstrual cycle of the participants should be considered to identify possible correlations between physical performance, dietary habits, and the hormonal system. However, for neither of the two biological sexes is the sample size sufficient to be able to make clear statements. However, no gender-specific difference could be observed in the individual courses either. Nevertheless, the results of this study provide important initial indications about athletically active individuals and their performance.

## 5. Conclusions

The aim of the study was to gather initial evidence on the acute effect of switching from an omnivorous to a vegan diet in the first 8 weeks. It has been shown that there were no changes in the upper and lower body performance. This is probably due to the high calorie intake combined with a high carbohydrate intake for both groups. In addition, a slight improvement in the body weight and BMI was observed. However, a significant reduction in the protein intake was observed during the vegan phase. The influence of these results on muscular adaptation and body composition cannot be assessed at this stage. In addition, no statement can be made about the distribution of amino acids or fatty acids during the two dietary phases as this was also not a part of this study. Therefore, these aspects should be taken into account in future studies.

## Figures and Tables

**Figure 1 ijerph-20-01856-f001:**
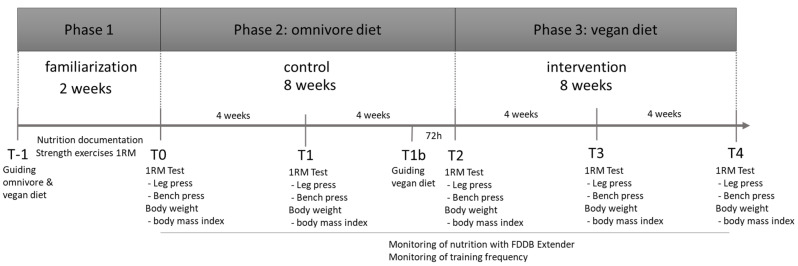
Study Design—presentation of the temporal course of the study. All 3 phases are shown with their measurement points. (1 RM = One repetition maximum test).

**Figure 2 ijerph-20-01856-f002:**
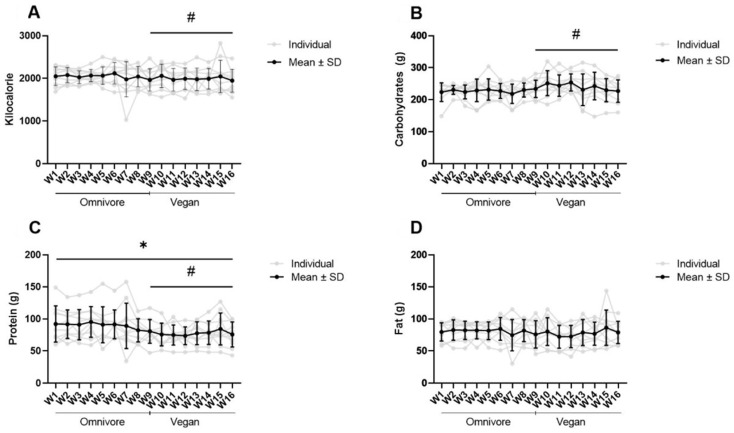
Change in self-documented calorie count (**A**) and macronutrients carbohydrates (**B**), proteins (**C**), and fats (**D**). All figures show the individual, self-documented intakes of the macronutrients or the calorie count in grey and the mean value in black. Significant differences were set *p* < 0.05 and market with # (group) and * (time).

**Table 1 ijerph-20-01856-t001:** Anthropometric Data.

	Omnivore Phase			Vegan Phase		Time Effect	Group Effect	Cohens d (Group)
	T0	T1	T2	T3	T4	T0–T4	W1-8 vs. W9-16	
Age [y]	29.30 ± 9.26					---	---	---
Height [m]	1.72 ± 0.08					---	---	---
Weight [kg]	68.92 ± 13.17	68.97 ± 13.17	69.13 ± 14.06	68.41 ± 13.70	67.82 ± 13.50	0.623	0.003 #	0.066
BMI [kg/m^2^]	23.20 ± 2.62	23.20 ± 2.79	23.25 ± 2.90	23.01 ± 2.75	22.81 ± 2.71	0.593	0.002 #	0.109

BMI = body mass index; significant differences were set *p* < 0.05 and market with # (group).

**Table 2 ijerph-20-01856-t002:** Change in strength capacity for the leg press and bench press.

	Omnivore Phase			Vegan Phase		Time Effect	Group Effect	Cohens d (Group)
	T0	T1	T2	T3	T4	T0–T4	W1-8 vs. W9-16	
LP [kg]	134.25 ± 59.81	136.75 ± 61.95	136.25 ± 64.32	137.75 ± 64.72	139.00 ± 67.77	0.242	0.368	---
LP (rel.)	1.89 ± 0.46	1.92 ± 0.45	1.91 ± 0.51	1.95 ± 0.53	1.98 ± 0.57	0.189	0.171	---
BP [kg]	52.25 ± 19.83	52.50 ± 19.14	53.25 ± 19.72	52.50 ± 18.35	52.75 ± 20.01	0.307	0.673	---
BP (rel.)	0.75 ± 0.21	0.75 ± 0.20	0.76 ± 0.20	0.76 ± 0.19	0.76 ± 0.20	0.431	0.229	---

BP = bench press; LP = leg press.

**Table 3 ijerph-20-01856-t003:** Relative protein intake.

	Omnivore Phase			Vegan Phase		Time Effect	Group Effect	Cohens d (Group)
	T0	T1	T2	T3	T4	T0–T4	W1-8 vs. W9-16	
Protein/Weight [g/kg]	1.37 ± 0.45	1.42 ± 0.38	1.22 ± 0.30	1.10 ± 0.22	1.13 ± 0.28	0.006 *	0.001 #	0.687

The table shows the development of the relative protein intake over the course of the study. Significant differences were set *p* < 0.05 and market with # (group) and * (time).

## Data Availability

The data presented in this study are available on request from the corresponding author.

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
