# Peer review of "Change to a Plant-Based Diet Has No Effect on Strength Performance in Trained Persons in the First 8 Weeks—A 16-Week Controlled Pilot Study"

_ijerph, 2023, doi:10.3390/ijerph20031856_

Round 1

Reviewer 1 Report

1) suggest need to explain why leg and bench press were chosen as assessment in this study

2) need to justify why 8 weeks was used in this study and whether is justify to see the changes 

3)Between 198 weeks 1 and 16, on average 103 ± 209kcal were added (Figure2A). However, a significant 199 difference with a small effect was noted between the two treatments (Mean: -60.65 SD: 200 98.47; p=.011#; d=.295): need to discuss the effect in long term

Author Response

Dear Reviewer,

Thank you for your constructive and important comments. Please find attached our point by point response to your comments and those of the other reviewers.

We hope that our revised version meets your requirements.

Reviewer 1

1) suggest need to explain why leg and bench press were chosen as assessment in this study

Answer:

Thank you very much for this comment. The leg press and chest press were chosen in this study because we wanted to check both lower body and upper body strength. We deliberately chose two exercises here that have an easy movement execution over one axis of movement. They are also exercises that are used by the majority of gym members. We also considered using the squat and the bench press instead of the two exercises, but deliberately decided against them due to the complexity of the movement. We could not guarantee that all participants performed these exercises regularly and might have obtained an adaptation effect based on the repetition of the measurement alone. We wanted to avoid this as much as possible.

2) need to justify why 8 weeks was used in this study and whether is justify to see the changes 

Answer:

Thank you also for this comment. We deliberately decided on the 8 weeks, as clear effects can usually be observed especially in the first weeks of a changeover. In addition, 8 weeks is the general length that ambitious recreational athletes first try out a change. If a change has a positive effect, it will of course be continued. If it has a negative effect, the athlete will revert to old habits (e.g. omnivorous diet). With this study, we wanted to develop a practical scenario that reflects for example a New Year's resolution and did not want to investigate the long-term effects of a vegan diet.

3) Between weeks 1 and 16, on average 103 ± 209kcal were added (Figure2A). However, a significant difference with a small effect was noted between the two treatments (Mean: -60.65 SD: 200 98.47; p=.011#; d=.295): need to discuss the effect in long term.

Answer:

Thank you very much for this constructive comment. We added a section in line 269.

Reviewer 2 Report

The aim of this pilot study is to investigate the effect of switching to a plant-based diet on the performance of strength-trained individuals.

I suggest to merge these two sentences : “Recent reviews have shown that the increased consumption of ani- 34 mal-based foods is referred to higher risks for various types of cancer [3,4]. Especially red 35 meat and highly processed products can increase the risk [5,6].

In line 59 you mentioned a well-planned vegan diet - what does that mean and who plans it?

In the introduction section, it is useful to mention some of the dietary components of a plant based diet (tannins, dietary fibre, phytic acid etc.) that affect the bioavailability of essential dietary components such as iron, zinc, proteins..

It would have been more professional to involve a sports dietitian/ dietitian in the study. How did the participants know, for example, how many grams of bread they ate for breakfast? - a more detailed description is needed

It's not clear how you assessed the plausibility of the energy intake?

It is not clear whether participants consumed dietary supplements such as protein powders, etc.. The self-documented diary is not reliable, as you noted in the limitations.

The methodological part needs to be described in more detail.

The calculation of BMR and energy expenditure is described in the methods, but the result is not given in the text.

In the references section, please check number 4.

Author Response

Dear Reviewer,

Thank you for your constructive and important comments. Please find attached our point by point response to your comments and those of the other reviewers.

We hope that our revised version meets your requirements.

Reviewer 2

the aim of this pilot study is to investigate the effect of switching to a plant-based diet on the performance of strength-trained individuals.

I suggest to merge these two sentences : “Recent reviews have shown that the increased consumption of animal-based foods is referred to higher risks for various types of cancer [3,4]. Especially red 35 meat and highly processed products can increase the risk [5,6].

Answer:

We merged these two sentences. Thank you for your help.

In line 59 you mentioned a well-planned vegan diet - what does that mean and who plans it?

Answer:

Thank you for this note. A well-planned vegan diet is meant here as a balanced diet. Furthermore, for athletes, this should be supervised by an expert such as a nutritionist.

We improved this section and add some more information. Line:69-71

In the introduction section, it is useful to mention some of the dietary components of a plant based diet (tannins, dietary fibre, phytic acid etc.) that affect the bioavailability of essential dietary components such as iron, zinc, proteins..

Answer:

Thank you for the important suggestion. We have added another section in the introduction regarding bioavailability and hope that you have targeted this. Line: 58-61

It would have been more professional to involve a sports dietitian/ dietitian in the study. How did the participants know, for example, how many grams of bread they ate for breakfast? - a more detailed description is needed

Answer:

Thank you also for this comment. The FDDB Explorer App is a valid method to monitor macronutrients and energy intake. In the app, you can enter grams as well as slices or products. Regarding supervision, we have used the wrong term here. The supervisor also has a sports science background. We have therefore adjusted the description.

It's not clear how you assessed the plausibility of the energy intake?

Answer:

Thank you for this important point. We only calculated the energy intake to give participants a guideline throughout the study. However, it was not our intention to set the participants on the theoretically calculated energy consumption and thus possibly already influence the performance or body weight. In our study, we were mainly interested in how the self-documented energy intake and the macronutrient distribution changed. It is of course well known that the actual energy expenditure cannot be calculated with a formula on a daily and situational basis. Moreover, in human studies it is not possible to check every single meal of the participants. Since we have this measurement inaccuracy in all phases, we believe that we can neglect it in this context for the time being.

It is not clear whether participants consumed dietary supplements such as protein powders, etc. The self-documented diary is not reliable, as you noted in the limitations.

Answer:

We added more information about the use of dietary supplements. We confirm with your comment about the reliability of the self-documented diary, but in human intervention studies there are limitations with any methodology. Nevertheless, the food diary is one of the most common methods in sports and fitness.

The methodological part needs to be described in more detail.

Answer:

Thank you very much for this comment. We added some more informations into the methodlogical part and hope that meets your requirements. Line: 111-117; 124; 139-144; 153-167.

The calculation of BMR and energy expenditure is described in the methods, but the result is not given in the text.

Answer:

Thank you for the comment. Yes, we did not include the calculated energy intake again as it was not explicitly different from the documented one and therefore had no added value for this study.

In the references section, please check number 4.

Answer:

Thank you, we checked our references.

Reviewer 3 Report

This manuscript aims to obtain initial findings on the influence of a diet change to veganism on the performance of strength-trained individuals. This is an interesting study. My comments are follows:

Abstract

L13: Remove ‘They hope…generation’

Introduction

L31: Provide the statistics to indicate the increasing no. of people abstaining/reducing the intake animal-based foods

L39: Provide full term of ‘CRP’

Methods

L84: Please indicate the location of the ‘local gym’

L85: How do you know the leg press & bench press are part of their training routine?

L161: State the equipment used to measure the height

Results

Good data presentation

Discussion

L243: Briefly elaborate why people tend to consume more carbohydrates during the vegan diet period. Any mechanism involves?

L260: How about the amino acid composition of the plant-based diet? A plant-based food might be high in protein content, but not a complementary protein.  

L274: For the limitations segment, please highlight that majority of the respondents is female.

Conclusion

Too short.

Author Response

Dear Reviewer,

Thank you for your constructive and important comments. Please find attached our point by point response to your comments and those of the other reviewers.

We hope that our revised version meets your requirements.

Reviewer 3:

This manuscript aims to obtain initial findings on the influence of a diet change to veganism on the performance of strength-trained individuals. This is an interesting study. My comments are follows:

Answer:

Thank you very much for this positive feedback. You will find our answer following your comments.

L13: Remove ‘They hope…generation’

Answer:

We improved this section and merged the two sentences.

L31: Provide the statistics to indicate the increasing no. of people abstaining/reducing the intake animal-based foods

Answer:

Thank you very much for this comment. We added some statistics to our statement

L39: Provide full term of ‘CRP’

Answer:

We changed to the full term.

L84: Please indicate the location of the ‘local gym’

Answer:

We added this information.

L85: How do you know the leg press & bench press are part of their training routine?

Answer:

Our sports nutritionist was also employed at the local gym and therefore knew all the participants and their training habits. This ensured that all participants knew the two exercises and performed them regularly. In addition, we were able to ensure that the training was not seriously changed during the entire period. We added more informations about the monitoring of the training habits.

Line: 111-117

L161: State the equipment used to measure the height

Answer:

Thank you for this note. We added the informations in line 196-197

Results

Good data presentation

Answer:

Thank you very much.

Discussion

L243: Briefly elaborate why people tend to consume more carbohydrates during the vegan diet period. Any mechanism involves?

Answer:

Vegan diets are completely devoid of animal products, so the missing calories and macronutrients have to be made up by plant foods. However, it is known that whole grains, legumes and seeds are rich in carbohydrates, which serve as the primary source of nutrition for vegans. As a result, the percentage of carbohydrates in total energy intake also increases.

We added some detailed information in our discussion. Line: 282-285.

L260: How about the amino acid composition of the plant-based diet? A plant-based food might be high in protein content, but not a complementary protein. 

Thank you for this important comment and we agree with you on this point. When comparing individual plant foods with animal foods, a low concentration of essential amino acids can be found in almost all plant foods (except soy and quinoa). However, a meal usually consists of at least two or three foods, not just one. As a result, the right combination (e.g. soy, potatoes, beans) can compensate for the missing amino acids. In addition, our focus in this pilot project was first on the general distribution of macronutrients in order to identify possible differences in athletically active people. However, we are also of the opinion that in future, in addition to the general protein intake, the amino acid composition must also be taken into account or analysed.

We added some detailed informations in our discussion. Line: 304-308.

L274: For the limitations segment, please highlight that majority of the respondents is female.

Answer:

Thank you very much for this important comment. We improved our limitation section and added some more informations. Line: 328-333

Conclusion - Too short.

Answer:

Thank you again. We improved our conclusion and hope it meets your requirement.